# Challenges in the Diagnosis of Tertiary Syphilis: Case Report with Literature Review

**DOI:** 10.3390/ijerph192416992

**Published:** 2022-12-17

**Authors:** Lucyna Jankowska, Zygmunt Adamski, Adriana Polańska, Monika Bowszyc-Dmochowska, Katarzyna Plagens-Rotman, Piotr Merks, Magdalena Czarnecka-Operacz, Ryszard Żaba

**Affiliations:** 1Department of Dermatology, Heliodor Święcicki Clinical Hospital, Poznan University of Medical Sciences, 60-806 Poznan, Poland; 2Department of Dermatology and Venenerology, Poznan University of Medical Sciences, 60-806 Poznan, Poland; 3Center for Pediatric, Adolescent Gynecology and Sexology Division of Gynecology, Department of Perinatology and Gynecology, Poznan University of Medical Sciences, 61-758 Poznan, Poland; 4Department of Pharmacology and Clinical Pharmacology, Faculty of Medicine, Collegium Medicum, Cardinal Stefan Wyszynski University in Warsaw, 01-938 Warszawa, Poland; 5Allergic and Occupational Skin Diseases Unit, Department of Dermatology, Medical University of Poznań, 60-355 Poznan, Poland

**Keywords:** tertiary syphilis, diagnosis, treatment

## Abstract

Tertiary syphilis is a large diagnostic challenge. It is rarely the case that it affects the skin, bone tissue and the eyes at the same time. The presented case shows that extensive symptomatology of syphilis poses a challenge in making a proper diagnosis in patients whose history does not suspect STDs. The study aims to present the case of a young woman hospitalized with a suspected autoimmune disease, diagnosed with symptomatic late syphilis with involvement of the skin, bones and eyes.

## 1. Introduction

Syphilis is a systemic infectious disease caused by the gram-negative spirochete *Treponema pallidum*. Humans are its only natural host [1]. There are an estimated 6 million new syphilis cases each year worldwide in those aged 15–49 years [2]. According to the Polish National Institute of Public Health (PZH), 1008 new cases of syphilis were recorded in Poland in 2020, of which 29 were cases of congenital syphilis, 979 cases of early syphilis and 550 cases of late syphilis [3]. In recent years, the incidence of syphilis has been on the rise in Poland and globally, which is probably related to the higher number of risky sexual encounters with random people, especially among homosexual men (MSM—men who have sex with men) [4,5]. The most common way syphilis is spread is through sexual contact (sexual intercourse, kisses) when the bacteria enter the body through mucus membranes or broken skin [6]. A study published in the Journal of the American Medical Association (JAMA) in 1971 showed that during 30 days of observation the risk of becoming infected with syphilis through sexual contact with an infected person is 30% [7]. Syphilis can also be acquired via transplacental transmission. Non-sexual syphilitic infection is possible through contact with infected biological material, but *Treponema pallidum* does not contain lipopolysaccharides, which makes spirochetes very sensitive to external factors (drying, high temperature, antiseptics and detergents), which makes this way of transmission definitely less frequent [8]. Blood transfusions and transplants have a potential risk of transmitting syphilis, but that seldom occurs in developed countries [9].

Syphilis can be divided into primary and secondary stage early syphilis (within one year of infection), and late latent and symptomatic syphilis [10]. The number of cases of late symptomatic syphilis have dropped significantly since the implementation of penicillin to treat early syphilis [11], but occasional cases of tertiary syphilis are still noted [12,13,14,15,16,17,18,19,20,21,22,23,24,25,26,27,28,29,30,31,32,33,34,35,36,37,38,39,40,41,42,43,44,45,46,47,48,49,50,51,52,53,54,55,56,57,58,59,60,61,62,63,64,65,66,67,68,69,70,71,72,73,74,75,76,77,78]. Given the wide variety of presenting symptoms, its systemic character and long asymptomatic period, syphilis is also known as the great imitator. Below is a case report of a young woman hospitalised with suspected autoinflammatory disease, who was diagnosed with late syphilis affecting the skin, bones and eyes.

## 2. Materials and Methods

The medical database PubMed was searched for the MeSH terms “tertiary syphilis”, “late syphilis of the skin” and “tubero-serpiginous syphilis”. The inclusion criterion was articles published in a reviewed journal in the last 10 years (2012–2022). There were no limitations with regard to the publication language or the kind of study.

## 3. Case Report

A 20-year-old female patient was admitted to the Clinical Department of Dermatology with suspected pyoderma gangrenosum in the right scapular area, for extended diagnostics and qualification for general treatment with cyclosporine. The first skin lesions had appeared 9 months earlier, when the patient noticed an erythematous papule on the right scapula (Figure 1).

Over the next 3 months it gradually expanded and ulcerated with a thickened lump appearing in the centre (Figure 2).

The patient visited the Surgical Outpatient Clinic twice, where she was recommended an oral antibiotic therapy and underwent surgical treatment of the lesion, which did not improve the dermatological condition. The patient visited the University Dermatological Outpatient Clinic (Figure 3).

Bacteriology and mycology tests were negative. A biopsy specimen for histopathological examination was collected and topical treatment with betamethasone and gentamicin was applied. The histopathological assessment revealed an enlarged swollen epidermis, abundantly infiltrated with neutrophils and covered by an exudative-neutrophilic scab. Under the epidermis an abundant chronic inflammatory infiltration was found with a visible plasmocytic component. Clinical and pathological correlation for superficial granulomatous pyoderma gangrenosum or another chronic pyoderma was recommended (Figure 4).

The laboratory tests detected anti-nuclear antibodies (ANA) at a titre of 1:80.

The tests for hepatitis B and C virus and human immunodeficiency virus (HIV) were negative. Six months after skin lesions, the patient observed sudden deterioration in visual acuity and floaters in the field of vision. During hospitalisation in the Department of Ophthalmology, she was diagnosed with uveitis in both eyes. The patient’s dermatological condition worsened significantly in that period. The repeated test showed ANA at a titre of 1:1280. The Extractable Nuclear Antigen Antibodies panel (ENA) was negative. Because of the spread of skin lesions, dapsone was administered (100 mg/daily). In December 2021, during a rheumatological consultation due to spinal pain, a test of human leukocyte antigen B27 (HLA-B27) was performed (positive result). Ankylosing spondylitis was suspected and observation was recommended. After a two-month therapy with dapsone, the patient reported for an outpatient follow-up visit, which revealed enlargement of the skin lesion in the scapular area and an erythematous lesion covered with greasy scales on the scalp and in the right parietal area, as well as painful swelling in the middle part of the right clavicle. The patient was referred to the Department of Dermatology for extended diagnostics and treatment.

The history of chronic diseases was negative. During the physical examination, the patient reported erosions of the oral mucosa recurrent for 5 years and one episode of a painless erosion in the genital area about one year prior to hospitalisation, which resolved spontaneously after several weeks. She also complained about pain in the right tibia, the right hip joint and occasional periumbilical pain. The patient had been taking oral hormonal contraception for 3 years because of irregular menstruation. She did not report risky sexual behaviours during this history and had been having intercourse with her first sexual partner for 3 years. During hospitalization, the patient reported that over the past year her partner had noticed a painless lesion in the area of the penis, which resolved spontaneously without leaving a scar. Similarly, he denied other sexual contacts, claiming the patient has been his first and only sexual partner.

On admission in the Department of Dermatology, a painless single raised serpiginous erythematous-brown plaque (5 × 6 cm) was found with small ulcerations and central scarring on the right scapula (Figure 5).

On the scalp in the parietal area, erythema covered with greasy yellowish scales was observed, with slightly swollen skin and subcutaneous tissue (Figure 6).

A small healing erosion was found on the lower lip. On admission, nodular swelling of the right clavicle was found with a size of about a quail egg, with tenderness on palpation (Figure 7).

During hospitalisation, a histopathological examination was repeated. Direct and culture mycological examinations of skin lesions in the right scapular and right parietal areas were performed. No presence of fungi was found. Given the non-standard clinical course, lesions in the skin, eyes, bones and mucous membranes, Behçet’s disease was suspected. The patient was given 5 points according to the diagnostic criteria for this disease entity (International Criteria for Behçet’s Disease, ICBD) [79,80,81]. The results of the rheumatological and ophthalmological consultations also suggested systemic vasculitis in the course of Behçet’s disease. Apart from uveitis in both eyes and vitreous floaters, bilateral swelling of the optic discs was detected. Computed tomography and magnetic resonance angiography of the head excluded dural venous sinus thrombosis. Imaging diagnostics of the lesion in the right clavicle was performed (Figure 7B).

Computed tomography showed an osteolytic weakly delineated lesion with malignant periosteal reaction, swollen bones and damaged periosteum. The non-specific lesion in the clavicle required further diagnostics. Ewing’s sarcoma, eosinophilic granuloma and inflammatory lesions were suspected. The screening test results for syphilis were positive. The infection was confirmed by high values of serum reactions: rapid plasma reagin test (RPR)—1:256, *Treponema pallidum* hemagglutination assay test (TPHA)—1:20,480 and fluorescent treponemal antibody absorbent test (FTA-ABS)—1:3200. Then the patient’s partner was examined at the University Outpatient Clinic. An infection with syphilis was found with the following serum reactions: RPR—1:32, TPHA—1:10,240 and FTA-AABS—1:3200. Neither hepatitis B or C virus nor HIV were detected in the patient and her partner. The remaining laboratory tests revealed elevated levels of C-reactive protein (28.8 mg/L), alanine aminotransferase (35 U/L), alkaline phosphatase (111 U/L), D-dimer (0.66 μg/mL), total cholesterol (238 mg/dL), triglycerides (309 mg/dL) and calprotectin in stool (199 μg/g). A lower level of 25-hydroxycholecalciferol was noted (8 ng/mL). No significant deviations in blood count, renal parameters and tumour markers (CA 125, alpha-fetoprotein, carcino-embryonic antigen, CA 19.9) were observed.

The diagnosed infection with syphilis in the patient radically changed the treatment. Due to the suspicion of neurosyphilis, the patient underwent a neurological consultation. The neurological examination showed a round pupil with normal reaction to light in the right eye and maintained consensual reaction, with an irregular wide and non-responsive pupil in the left eye. CSF-TPHA test results were positive. The remaining parameters of the cerebrospinal fluid were: cytosis (1/μL), erythrocytes (2/mL), glucose (56 mg/dL), protein (183 mg/L).

The patient underwent an orthopaedic consultation. Based on the clinical picture, laboratory and imaging diagnostics, the orthopaedist suspected congenital syphilis with Higoumenakis’ sign. A core-needle biopsy revealed a non-specific picture of fragments of the trabecular bone surrounded by chronic granulomatous partially responsive inflammation, excluding tumour growth. An examination for syphilis in the patient’s parents did not confirm any infection.

A histopathological examination of the skin lesion performed during hospitalisation revealed a thickened non-regular epidermis with streaky swelling, thinned by lymphocytic and histiocytic inflammatory infiltration with single plasmacytes. Slightly thicker infiltration presented in the area of enlarged vessels of the superficial plexus. There was a lack of characteristics typical for active pyoderma gangrenosum. This clinical picture can be consistent with secondary/tertiary syphilis (Figure 8 and Figure 9).

A revision of the biopsy from September 2021 showed a thickened epidermis and lymphocytic and histiocytic inflammatory infiltration with multiple plasmacytes (Figure 10 and Figure 11).

Based on the clinical picture, numerous laboratory and imaging examinations, as well as consultations, tertiary syphilis was diagnosed in the skin, central nervous system, eyes and bones. During hospitalisation, crystalline penicillin (4 × 5 million units i.v.) was administered for 14 days, which improved the inflammatory parameters, the dermatological lesion in the right clavicle and the lesions on the scalp, as well as reducing the pain in the joints. On the last day of hospitalisation, the patient was given 2.4 million units of benzathine benzylpenicillin i.m. In the first days of treatment with crystalline penicillin, the patient also received methylprednisolone (24 mg p.o.) in order to avoid Jarisch Herxheimer reaction. The patient remains under the care of Dermatological, Orthopaedic, Neurology and Ophthalmology Outpatient Clinics. During the outpatient follow-up in May 2022, a total resolution of the skin lesions on the scalp and healing of the ulceration in the right clavicle leaving a mosaic scar were noted (Figure 12 and Figure 13). A slightly reduced tumour in the shaft of the right clavicle was observed (Figure 14). Serum examinations conducted in September 2022 showed RPR reaction at the titre of 1:4.

## 4. Discussion

Late symptomatic syphilis is a rare systemic disease that can affect the skin, the mucous membranes, the circulatory system, the nervous system and the skeletal system. The incidence of tertiary syphilis decreases significantly with the introduction of penicillin in its early stage, but new cases of this disease are still being reported [12,13,14,15,16,17,18,19,20,21,22,23,24,25,26,27,28,29,30,31,32,33,34,35,36,37,38,39,40,41,42,43,44,45,46,47,48,49,50,51,52,53,54,55,56,57,58,59,60,61,62,63,64,65,66,67,68,69,70,71,72,73,74,75,76,77,78,79]. It is estimated that about 25% of untreated patients will develop tertiary syphilis [82]. A conviction about the casuistic occurrence of late symptomatic syphilis does not allow doctors to take it into account in a differential diagnosis of non-standard symptoms. Lesions in tertiary syphilis are not infectious and contain a practically undetectable number of spirochetes, while serum reactions are highly positive. One person can develop more than two forms of this disease. Skin lesions in tertiary syphilis are tubero-serpiginous syphilis, tubero-ulcerative syphilis and gummatous syphilis. The lesions are usually single, painless, spreading peripherally with destruction of associating tissues [8,83]. As in the case described, they can imitate lesions in pyoderma gangrenosum, granuloma annulare, sarcoidosis, discoid lupus erythematosus, and even psoriasis [9,10,11,13,16].

Depending on the depth and intensity of necrosis, skin lesions can appear as superficial skin nodules (tubero-serpiginous syphilis) or deep subcutaneous nodules and gummas. In tubero-serpiginous syphilis, nodules are located in the dermis, while skin gummas concern subcutaneous tissue and deeper layers of the dermis, with intensified necrotic lesions. Both of the forms are characterised by a chronic (months-long or multi-year) course with a tendency to ulcerations and peripheral spread with spontaneous resolving in the middle part, leaving a scar.

In tubero-serpiginous syphilis the primary lesion is a nodule (from a few millimetres to a few centimetres), characterised by considerable cohesion and a bluish-brown colour. Over time, the nodule disintegrates in the central part, leaving a mosaic scar. New nodules appear on the periphery of the spreading tubero-serpiginous syphilis. Unevenly enlarging lesions often take a kidney-like shape (spread in a serpiginous manner). Active lesions have a shaft of confluent nodules around the periphery of the scar [83].

The histological picture in late syphilis is characterised by a proliferative reaction with a tendency for necrosis and perivascular infiltrations of lymphocytes, histiocytes and plasma cells [83].

In differentiating skin lesions, pyoderma gangrenosum and superficial granulomatous pyoderma (SGP) were taken into consideration. Lesions in pyoderma gangrenosum are painful, but in the case of superficial granulomatous pyoderma ulcerations are painless. In early and late syphilis, skin lesions are painless.

The histopathological picture of pyoderma gangrenosum is non-specific. In active untreated lesions, neutrophil infiltrations with leucocytoclasis are usually observed.

The histopathological picture of superficial granulomatous pyoderma shows a three-layer granulomatous inflammatory infiltration with central narcosis and neutrophils, surrounded by a ring of plasma cells and histiocytes, as well as the outermost ring of plasma cells and eosinophiles [84].

The plasma cells in the inflammatory infiltration in the histopathological picture of the patient did not exclude SGP, yet it is more typical for syphilis lesions. The correlation of the histopathological image with the clinical image and positive serological reactions for syphilis permitted diagnosis.

In 2018 Schliemann et al. documented a case of tubero-ulcero-serpiginous syphilis in a 44-year-old man with acquired immune deficiency, with ulcerating reddish-brown plaques on his arm. Neurosyphilis was ruled out and the patient was treated with three intramuscular injections of benzathine benzylpenicillin (2.4 mln units) at weekly intervals [18].

In our patient, lesions of similar morphology were observed, but due to the involvement of the eyes and suspicion of syphilis of the CNS, crystalline penicillin and gluco-corticosteroids were administered, in accordance with the 2018 recommendations of the Polish Dermatological Association and the 2020 European guidelines for the treatment of syphilis.

Bone lesions in late syphilis appear as osteomyelitis often accompanied by periosteal reaction, which leads to thinning foci surrounded by hypercalcified areas. The inflammatory reaction is usually of a proliferative character, causing a nodular rise of the bone with a periosteal reaction. In some cases, disintegration of periosteal exostoses takes place.

In tertiary syphilis, the skeletal system may also develop gummas with signs of disintegration and thinning of the bone structure, as well as proliferative changes and formation of new bone tissue in the periphery. Gummas can lead to punctured bones with a release of necrotic tissues and ulceration within the adjacent skin [83].

Bone lesions in tertiary syphilis are currently rarely described [41,42,43,44,45,46,47,48,49,50,51]. A group of researchers led by Wang presented a case of a 47-year-old woman with a gumma in the vertebral body of the thoracic spine and positive serum reactions, primarily suspected of a proliferative process. The biopsy revealed a granulomatous inflammatory reaction [43]. The histopathological examination showed a granulomatous and partially resorptive inflammatory process.

Late syphilis involving the internal organs (Lues visceralis) most often affects the cardiovascular system [52,53,54,55,56,57,58,59,60,61,62,63,64,65,66,67,68,69,70,71,72], mainly as aortitis that can develop aneurysms. Gummas can also be located in the lungs, the liver, the pancreas, the digestive tract and even the adrenal glands [73,74,75,76,77]. Tee Sa et al. described an extremely rare case of tertiary syphilis with gummas in the adrenal glands that led to their insufficiency [77].

Landry et al. carried out a retrospective study in Virginia, in which they analysed cases of neurosyphilis and tertiary syphilis between 1973–2017. Of 8874 cases of syphilis diagnosed at that time, 251 were neurosyphilis, the were syphilis of the cardiovascular system, and no case of gummatous syphilis was noted. The most frequent manifestation of neurosyphilis was the involvement of the eyes [78].

Neurosyphilis can occur at any stage of infection with *Treponema pallidum* (both in early syphilis and later), as spirochaetes penetrate the nervous system very early.

Bacteria in cerebrospinal fluid were detected 3 weeks after infection. The symptoms of the involvement of the central nervous system are usually not very characteristic [83,85,86,87], e.g., asymptomatic meningitis, acute syphilitic meningitis, meningo-vascular syphilis and syphilis of the spinal cord, gummas of the central nervous system, progressive palsy, tabes dorsalis. Neurosyphilis can be characterised by poor neurological and/or psychiatric symptoms [87].

Ocular syphilis is usually associated with early neurosyphilis with acute meningitis [79], and is manifested by uveitis, which can develop in secondary and tertiary syphilis.

Cases of an affected optical nerve with a swollen optic disc were also described. In 2022, Eijmael et al. reported a case of a 53-year-old woman with the symptoms of deterioration of vision and a swollen optical disc in the left eye. Initially, a giant cell arteritis was suspected, but extended diagnostics detected a tertiary stage ocular infection of syphilis in the form of uveitis. The patient was treated with high-dose of intravenous benzyl penicillin (24 million units) over 14 days [88].

The frequency of late syphilis of the skin is not fully known. In order to illustrate this issue, the PubMed database was searched for cases of tertiary syphilis in the last 10 years. 22 cases of late skin syphilis (including one in Poland) [14,18,19,20,21,22,23,24,25,26,27,28,29,30,31,32,33,34,35,36,37,47], 21 cases of cardiovascular system syphilis [52,53,54,55,56,57,58,59,60,61,62,63,64,65,66,67,68,69,70,71,72], five cases of systemic syphilis (the liver, the pancreas, the adrenal glands) [73,74,75,76,77], 11 cases of syphilis affecting bones [41,42,43,44,45,46,47,48,49,50,51], 17 cases of neurosyphilis [24,34,47,49,84,89,90,91,92,93,94,95,96,97,98,99,100], and one case of congenital late syphilis were found [89] (Table 1).

An interesting case of a female patient was presented. Her initial dermatological picture, the result of a histopathological examination and outpatient observation for ankylosing spondylitis pointed towards autoinflammatory disease affecting the skin. New clinical data of recurrent uveitis in both eyes, recurrent erosions and ulcerations in the oral cavity led to a differential diagnosis for Behçet’s disease. Initially, the lesion on the clavicle was suspected to be a malignant neoplastic process. A differential diagnosis for cutaneous tuberculosis was performed. A negative Quantiferon TB Gold was obtained.

Routine screening for syphilis turned out to be crucial for proper diagnosis and therapeutic treatment. By suspecting an autoimmune disease with a high level of antinuclear antibodies and making observations for the proliferative process, the result of the biological screening test was initially considered to be false reaction. The results of treponemal and non-treponemal serum reactions and positive results of the patient’s sexual partner ultimately confirmed the diagnosis. Both the patient and her infected partner did not mention risky sexual behaviours in the history, only kisses between 14–17 years of age. The manner of becoming infected has remained unexplained.

Congenital late syphilis suggested by the consulting orthopaedist was excluded due to negative serological reaction of the parents.

Due to a systemic course of the disease, the patient required multiple specialist consultations (neurological, gastroenterological, ophthalmological, rheumatological, psychological). During the neurological consultation, due to the suspicion of neurosyphilis, cerebrospinal fluid was collected, which revealed positive TPHA reaction. The examination of VDRL reaction in cerebrospinal fluid was not possible for organisational reasons. Despite normal levels of cytosis, protein and glucose, due to the several-month-long uveitis and swelling of the lenses in both eyes, it was decided to administer crystalline penicillin intravenously, in accordance with the 2018 diagnostic and therapeutic recommendations of the Polish Dermatological Association. To avoid Jarisch Herxeimer reaction, the patient received glucocorticosteroid, in accordance with the 2020 European guidelines for the treatment of syphilis.

During hospitalisation and outpatient observations, improved dermatological condition, healing of the ulceration, resolution of infiltration in the right clavicle leaving a mosaic scar, and complete resolution of inflammatory lesions on the scalp were noted. Serum reactions verified one month and three months after completion of the treatment did not show a decrease in the titre and the test carried out in September 2022 revealed RPR had reduced by 1:4. The patient remains under the care of dermatological, neurological, ophthalmological and orthopaedical outpatient clinics.

The patient’s partner had positive serological reactions for syphilis and was admitted to the Department of Dermatology. Extended diagnosis ruled out involvement of the CNS. The patient did not report pain and no significant deviations in laboratory test results and basic imaging examinations were observed. On the basis of the clinical picture, syphilis of unknown duration was diagnosed and the patient was treated with three intramuscular injections of bezanthine benzylpenicillin (2.4 mln units) at weekly intervals.

## 5. Conclusions

Tertiary syphilis is a large diagnostic challenge. It is rarely the case that it affects the skin, bone tissue and the eyes at the same time. The presented case shows that extensive symptomatology of syphilis poses a challenge in making a proper diagnosis in patients whose history does not suspect STDs. Given the increase in infections with *Treponema pallidum* over the last years in Poland and worldwide, screening tests should be taken into account in a differential diagnosis of dermatological, ophthalmological and neurological disorders.

## Figures and Tables

**Figure 1 ijerph-19-16992-f001:**
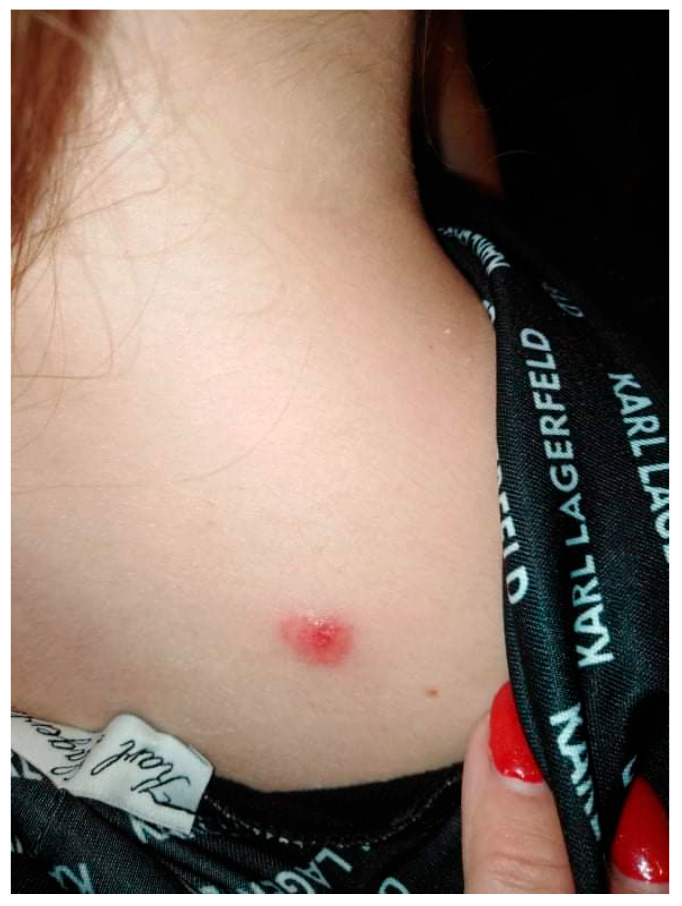
Erythematous papule on the right scapula.

**Figure 2 ijerph-19-16992-f002:**
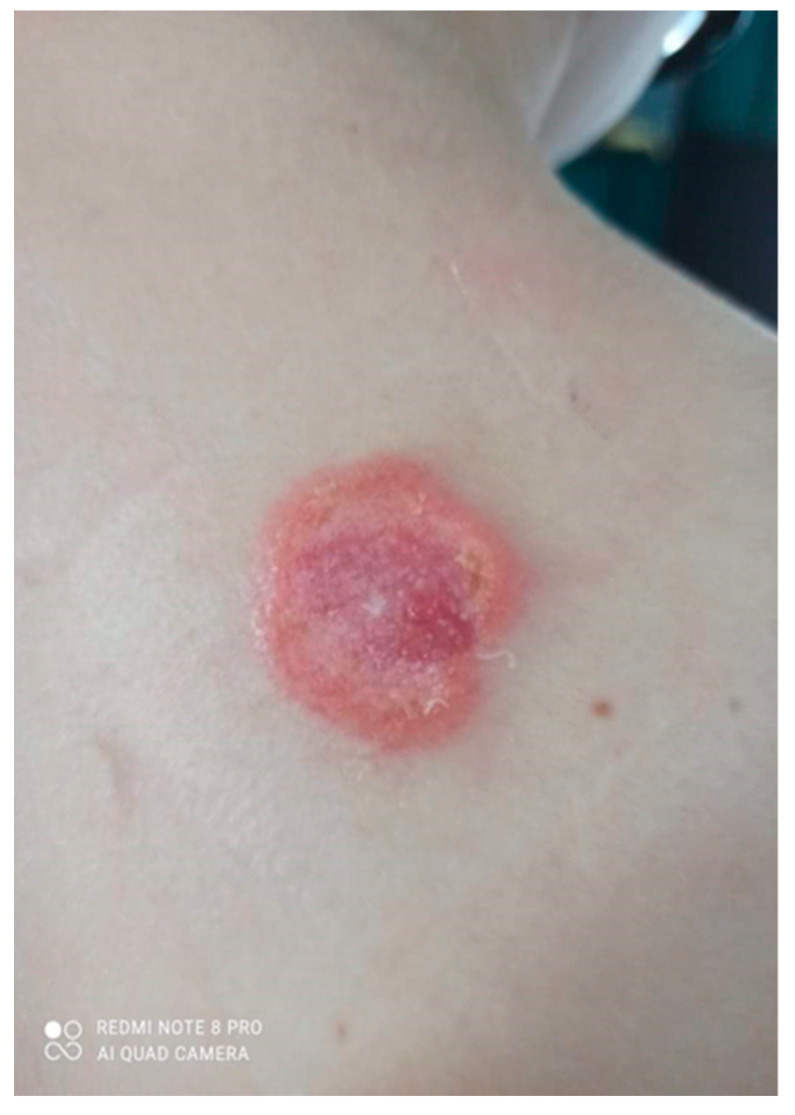
Gradually growing lesion with a thickened lump, the beginning of disintegration of the central papule.

**Figure 3 ijerph-19-16992-f003:**
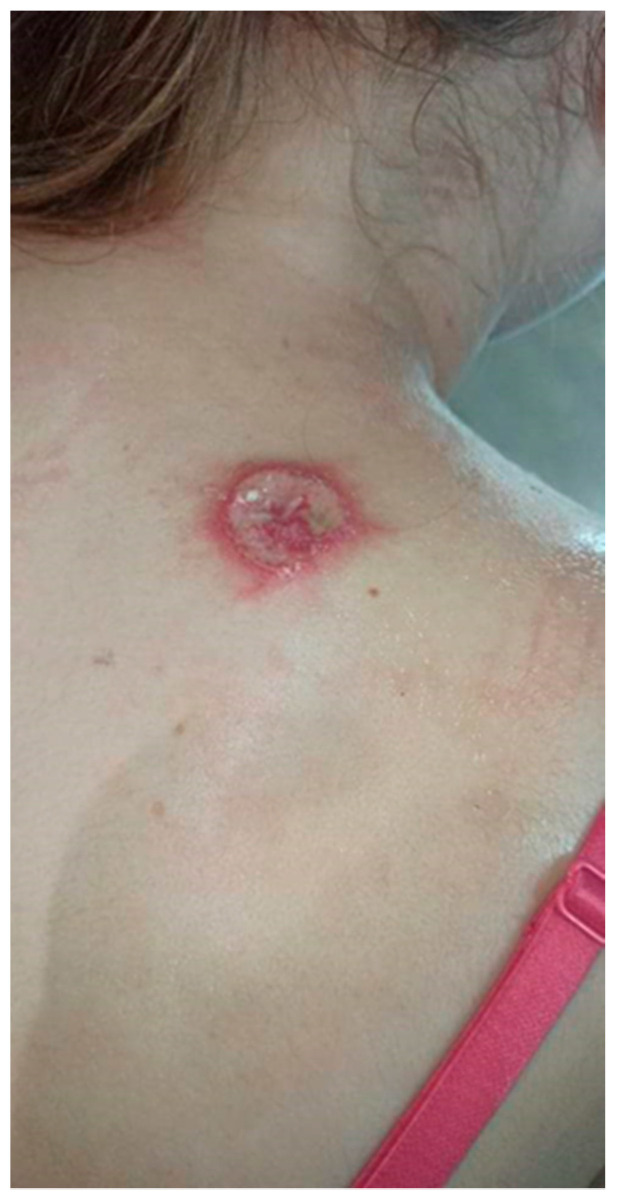
Ulceration with an active edge.

**Figure 4 ijerph-19-16992-f004:**
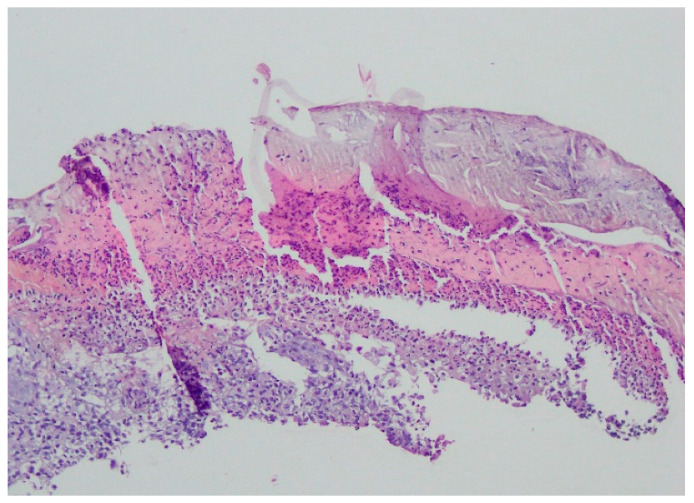
Enlarged swollen epidermis, abundantly infiltrated with neutrophils and covered by an exudative-neutrophilic scab. Under the epidermis, an abundant chronic inflammatory infiltration was found with a visible plasmocytic component. Clinical and pathological correlation for superficial granulomatous pyoderma gangrenosum or another chronic pyoderma was recommended.

**Figure 5 ijerph-19-16992-f005:**
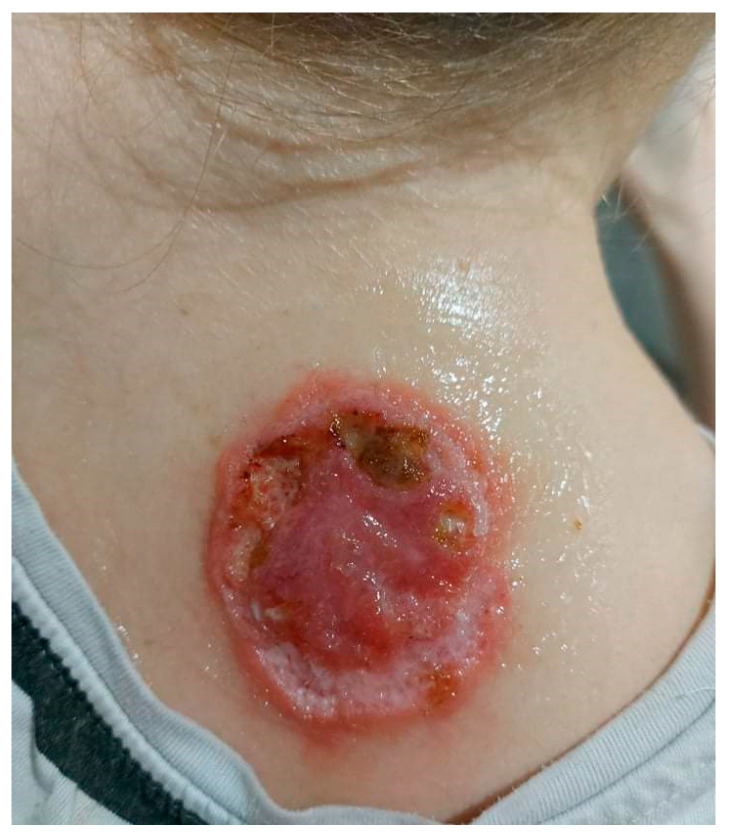
On admission in the Department of Dermatology, a single raised serpiginous erythematous-brown plaque (5 × 6 cm) with small ulcerations and the central scarring on the right scapula.

**Figure 6 ijerph-19-16992-f006:**
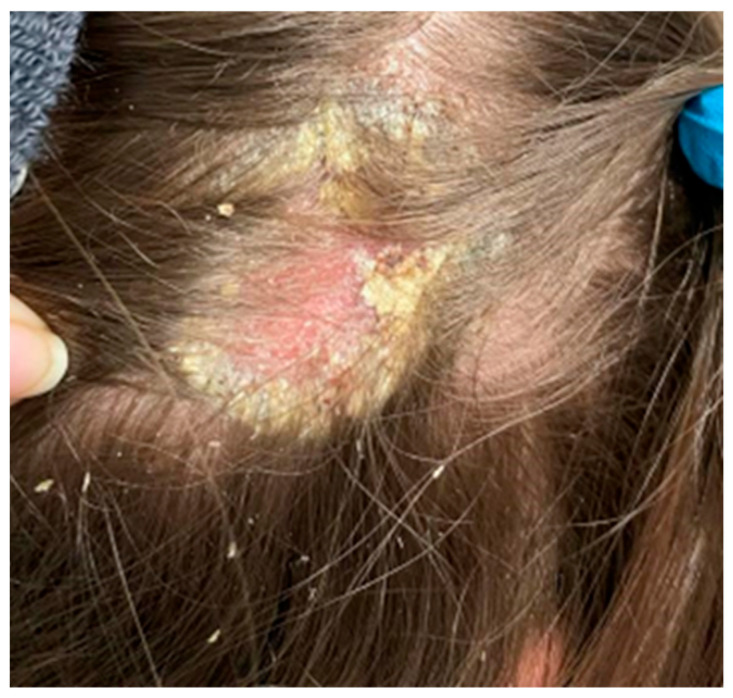
In the right parietal area, erythema covered with greasy yellowish scales, with swollen skin and subcutaneous tissue.

**Figure 7 ijerph-19-16992-f007:**
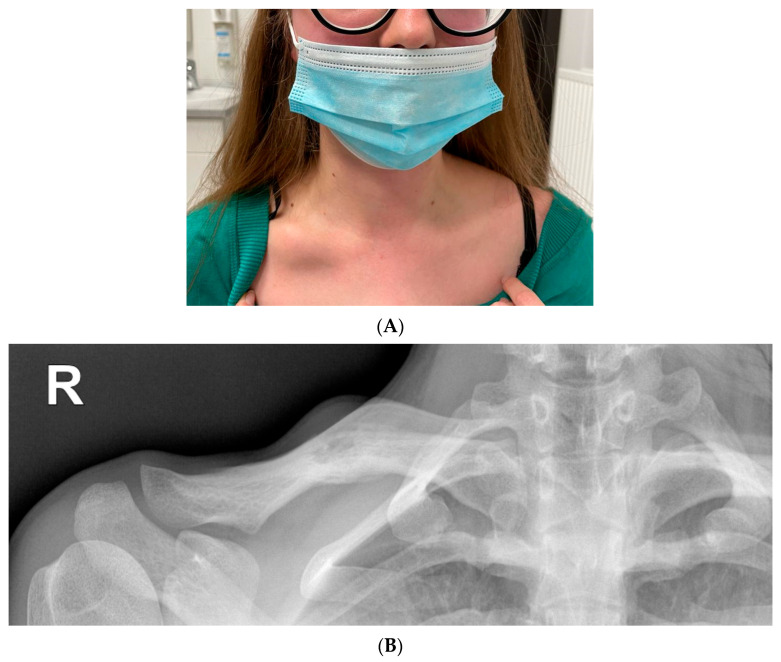
(**A**) Lumpy distension of the right clavicle the size of a quail egg with accompanying palpation tenderness. (**B**) Radiolucency on the right clavicle, probably an osteolytic lesion (15 mm). Linear shading along the bottom edge of the clavicle—periosteal reaction. Extended diagnostics recommended.

**Figure 8 ijerph-19-16992-f008:**
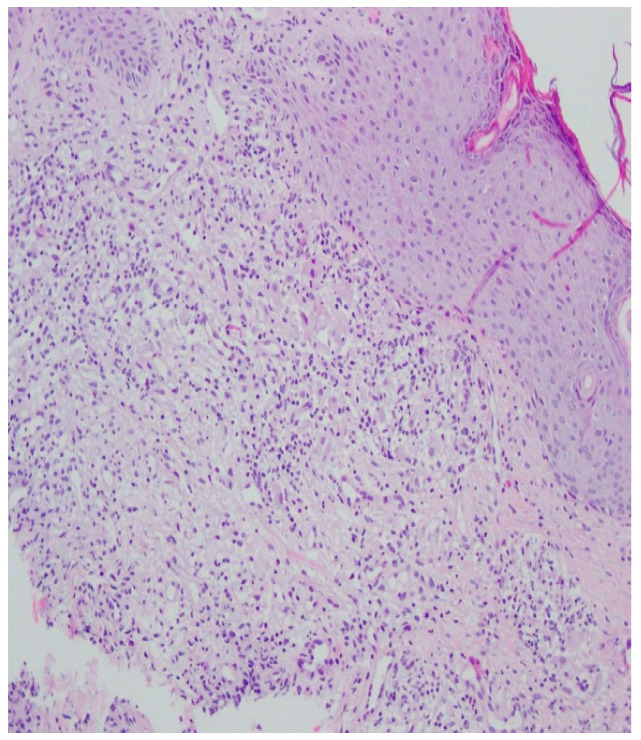
Thickened non-regular epidermis with streaky swelling, thinned by lymphocytic and histiocytic inflammatory infiltration with single plasmacytes. Slightly thicker infiltration in the area of enlarged vessels of the superficial plexus. Lack of characteristics typical for active pyoderma gangrenosum. The clinical picture can be consistent with secondary/tertiary syphilis.

**Figure 9 ijerph-19-16992-f009:**
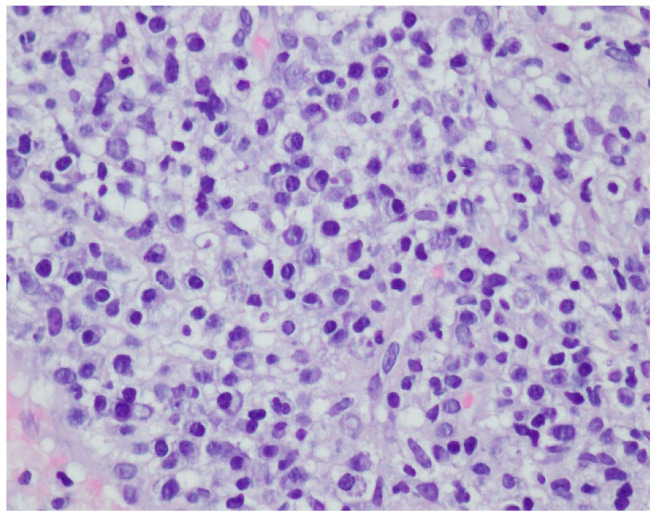
Thickened non-regular epidermis with streaky swelling, thinned by lymphocytic and histiocytic inflammatory infiltration with single plasmacytes. Slightly thicker infiltration in the area of enlarged vessels of the superficial plexus. Lack of characteristics typical for active pyoderma gangrenosum. The clinical picture can be consistent with secondary/tertiary syphilis.

**Figure 10 ijerph-19-16992-f010:**
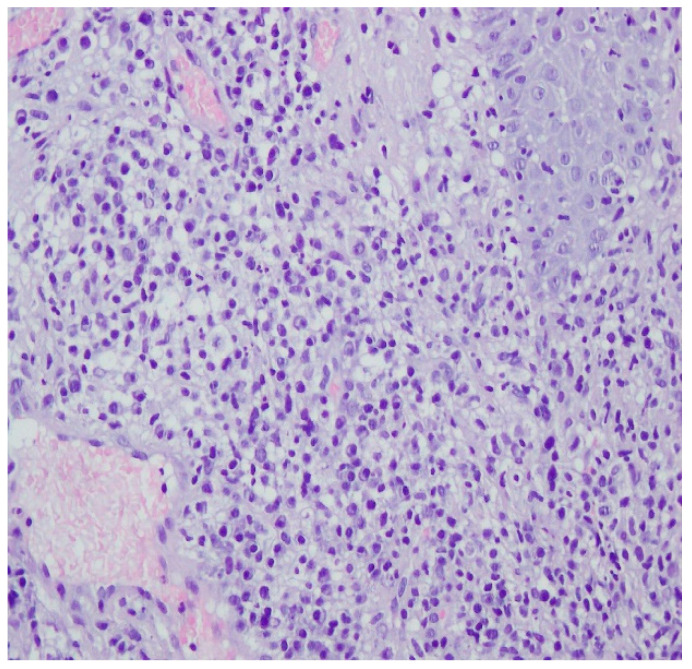
Thickened epidermis. Lymphocytic and histiocytic inflammatory infiltration with multiple plasmacytes. Histopathological picture and data from the history are consistent with syphilis.

**Figure 11 ijerph-19-16992-f011:**
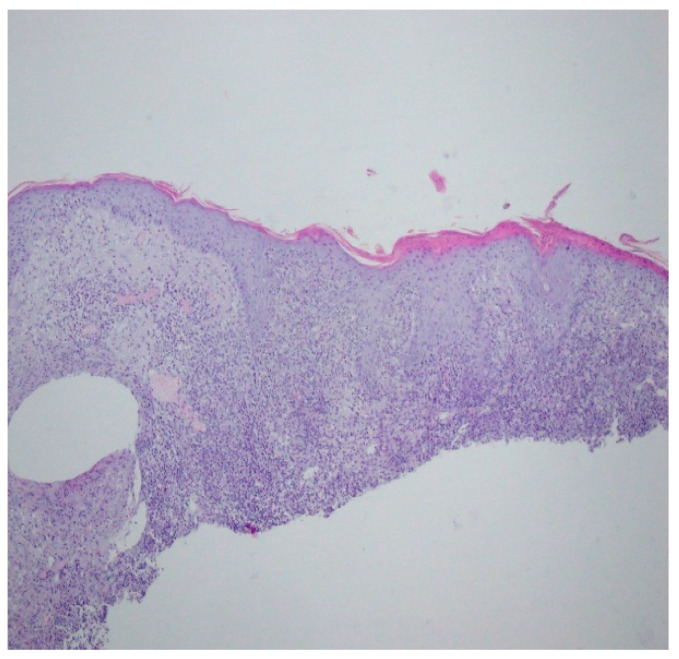
Thickened epidermis. Lymphocytic and histiocytic inflammatory infiltration with multiple plasmacytes. Histopathological picture and data from the history are consistent with syphilis.

**Figure 12 ijerph-19-16992-f012:**
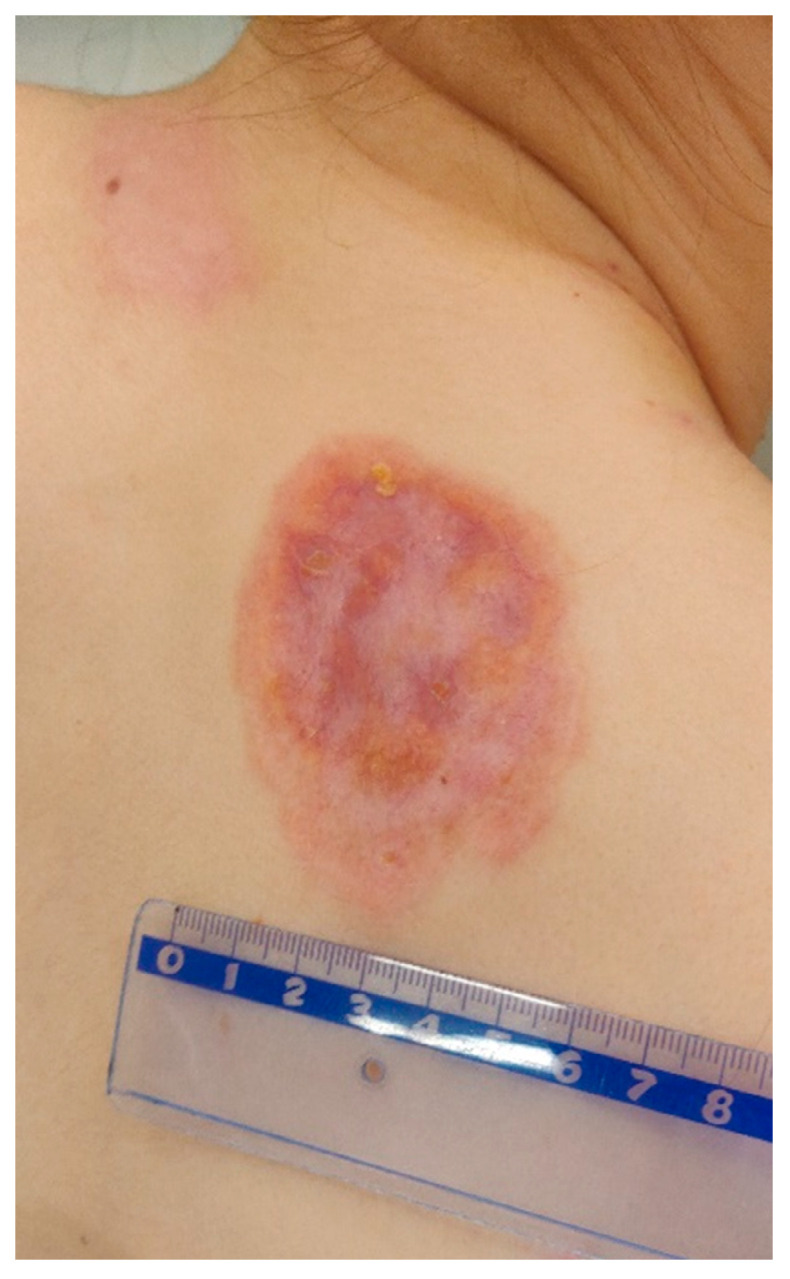
Lesions examined during the follow-up at the University Dermatological Outpatient Clinic (6 months after hospitalisation).

**Figure 13 ijerph-19-16992-f013:**
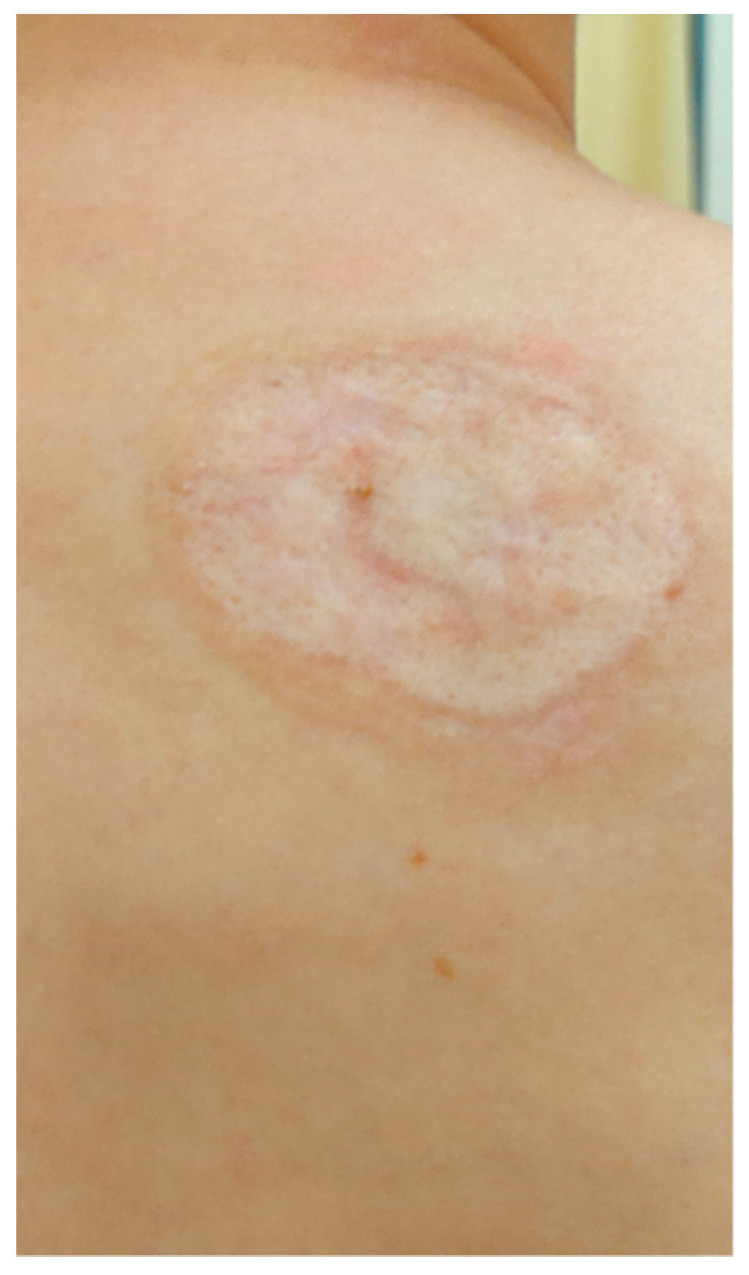
Lesions examined during the follow-up at the University Dermatological Outpatient Clinic (6 months after hospitalisation).

**Figure 14 ijerph-19-16992-f014:**
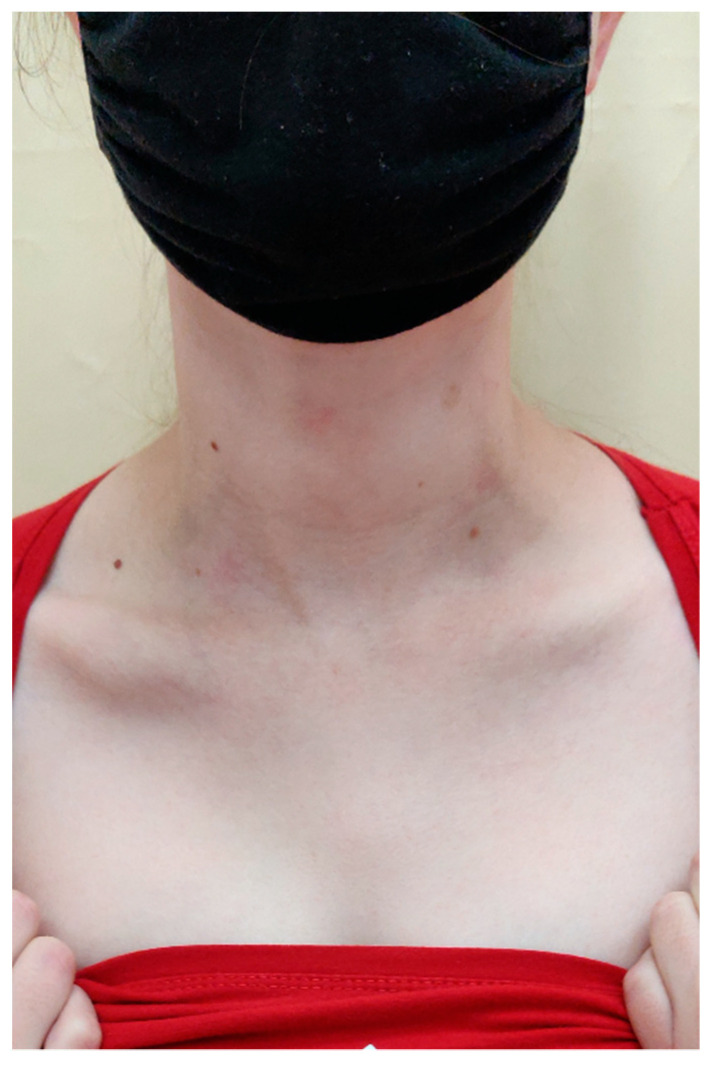
The right scapula on the last day of hospitalisation.

**Table 1 ijerph-19-16992-t001:** Number of case reports of tertiary syphilis based on the PubMed database from the last 10 years.

Type of Late Syphilis	Tubero-Serpiginous Syphilis, Tubero-Ulcerative Syphilis and Gummatous Syphilis	Cardiovascular System Syphilis	Lues Visceralis	Lues Tarda Ossium	Lues Nervosa	Lues Congenita Tarda
The number of cases of syphilis in 2012–2022	22	21	5	11	17	1

## Data Availability

Department of Dermatology, Heliodor Święcicki Clinical Hospital of the Poznan University of Medical Sciences, Poznan, Poland.

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
