# Peer review of "Challenges in the Diagnosis of Tertiary Syphilis: Case Report with Literature Review"

_ijerph, 2022, doi:10.3390/ijerph192416992_

Round 1

Reviewer 1 Report

This article described an unusual case of tertiary syphilis affecting skin, bone and eyes. The authors also reviewed literature.

This manuscript is well written.  The diagnosis of tertiary syphilis is very challenging because it has a wide variety of clinical presentations, and may occur decades after infection.  This case presentation with literature review is valuable for early diagnosis and treatment of this disease clinically.

For figure 4 and figure 11, please provide additional photos taken at high power view (400x) to show components of the chronic inflammatory infiltrate and the presence of plasma cells.  Plasma cells are commonly seen in syphilis, but the histologic features seen in these biopsies (figure 2, 8, 9, 10, 11) are nonspecific and not fully diagnostic unless spirochetes can be highlighted by immunostains.  It will be interesting to know if immunostains for T pallidum can be performed in these biopsies retrospectively although the sensitivity may not be high in tertiary syphilis.

Author Response

Dear Reviewer

I have an additional picture for Figure 11 (magnification 400 times) with visible plasmocytes. Unfortunately, I do not have a magnification for Figure 4. The immunohistochemical examination demonstrating the presence of spirochetes in lesion specimens is not available at our facility and we are not able to perform this retrospectively. Although the histopathological picture is not pathognomonic for syphilis, its correlation with the clinical picture and positive serological reactions for syphilis permitted diagnosis.

With best and warmest regards,

Authors      

Reviewer 2 Report

Dear authors,

I read your manuscript with great interest. Well-written and easy to follow.

Very interesting patient case of tertiarry syphilis.

Following comments are to be answered/discussed:

- The skin lesions of syphilis were painless. Granuloma pyogenicum lesions are always painful and the pain is a pathognomonic sign of this disease, but this was not the case in this patient. This issue should be discussed.

- Presence of plamacytes is contradictory to initial diagnosis of granuloma pyogenicum, but typical for syphilis. This issue should be also explained.

- Orthopedic diagnosis was congenital syphilis with higoumenakis sign. But parents had negtive syphilis serology. It would be interesting to explain whether authors are with the diagnosis of congenital syphilis or not.

- There is no information whether the partner of the patient has been also treated or not, if yes, which treatment exactly and how was the follow up.

Author Response

Dear Reviewer

In differentiating skin lesions, pyoderma gangrenosum and superficial granulomatous pyoderma (SGP) were taken into consideration. Lesions in pyoderma gangrenosum are painful, but in the case of superficial granulomatous pyoderma, ulcerations are painless. In early and late syphilis, skin lesions are painless.

The histopathological picture of pyoderma gangrenosum is non-specific. In active untreated lesions, neutrophil infiltrations with leucocytoclasis are usually observed.

The histopathological picture of superficial granulomatous pyoderma shows a three-layer granulomatous inflammatory infiltration with central narcosis and neutrophils, surrounded by a ring of plasma cells and histiocytes, as well as the most outer ring of plasma cells and eosinophiles.

The plasma cells in the inflammatory infiltration in the histopathological picture did not exclude SGP, yet plasma cells are more typical for the histopathological picture of syphilis lesions. 

Histological lesions in late syphilis are characterised by a proliferative reaction with a tendency for necrosis and perivascular infiltrations of lymphocytes, histiocytes and plasma cells.

The consulting orthopaedist diagnosed congenital late syphilis, which had been ruled out by our team due to negative biological reaction of the parents. Bone lesions of the patient may be observed in tertiary syphilis. 

The patient’s partner had positive serological reactions for syphilis and was admitted to the Department of Dermatology. Extended diagnosis ruled out involvement of the CNS. The patient did not report pain and no significant deviations in laboratory test results and basic imaging examinations were observed. On the basis of the clinical picture, syphilis of unknown duration was diagnosed and the patient was treated with three intramascular injections of bezanthine benzylpenicillin (2.4 mln units) at weekly intervals

With best and warmest regards,

Authors      

Reviewer 3 Report

Summary

This case study reported a clinical case of tertiary syphilis and conducted a literature review. It is interesting and important to the field of syphilis prevention and treatment, and can be helpful for the future works. Overall I think this case report had a high quality. I had no major comment, but I still have some minor comments:

Minor Issues

1.     I would suggest moving the PubMed searching strategies from lines 285-289 to the Methods section.

2.     More recommendations on future clinical practices, including diagnosis and prevention strategies can be needed in the discussion.

Author Response

Dear Reviewer

The table is included in the material and methods.

The discussion was supplemented with:

In our patient, lesions of similar morphology were observed, but due to the involvement of the eyes and suspicion of syphilis of the CNS, crystalline penicillin and glucocorticosteriod were administered, in accordance with the 2018 recommendations of the Polish Dermatological Association and the 2020European guidelines for the treament of syphilis.

Despite normal levels of cytosis, protein and glucose, due to the several-month-long uvei-tis and swelling of the lenses in both eyes, it was decided to administer crystalline penicil-lin intravenously, in accordance with the 2018 diagnostic and therapeutic recommenda-tions of the Polish Dermatological Association. To avoid Jarisch Herxeimerreaction, the patient received glucocorticosteroid, in accordance with the 2020 European guidelines for the treatment of syphilis.

With best and warmest regards,

Authors      
